# Ex Vivo Immune Responsiveness to SARS-CoV-2 Omicron BA.5.1 Following Vaccination with Unmodified mRNA-Vaccine

**DOI:** 10.3390/vaccines11030598

**Published:** 2023-03-06

**Authors:** Anna Sabrina Kuechler, Eva Heger, Maike Wirtz, Sandra Weinhold, Markus Uhrberg, Fritz Boege, Karin Schulze-Bosse

**Affiliations:** 1Central Institute for Clinical Chemistry and Laboratory Diagnostics, Medical Faculty, University Hospital Düsseldorf, Heinrich-Heine-University, 40225 Düsseldorf, Germany; 2Institute for Virology, Faculty of Medicine and University Hospital Cologne, University of Cologne, 50935 Cologne, Germany; 3Institute for Transplantation Diagnostics and Cell Therapeutics, University Hospital Düsseldorf, Heinrich-Heine-University, 40225 Düsseldorf, Germany

**Keywords:** COVID 19-serology, SARS-CoV-2-neutralization, SARS-CoV-2-vaccination, SARS-CoV-2-immunity, companion-diagnostic, SARS-CoV-2 BA.5.1

## Abstract

(1) Background: The high incidence of SARS-CoV-2 infection in vaccinated persons underscores the importance of individualized re-vaccination. PanIg antibodies that act against the S1/-receptor binding domain quantified in serum by a routine diagnostic test (ECLIA, Roche) can be used to gauge the individual ex vivo capacity of SARS-CoV-2 neutralization. However, that test is not adapted to mutations in the S1/-receptor binding domain, having accumulated in SARS-CoV-2 variants. Therefore, it might be unsuited to determine immune-reactivity against SARS-CoV-2 BA.5.1. (2) Method: To address this concern, we re-investigated sera obtained six months after second vaccinations with un-adapted mRNA vaccine Spikevax (Moderna). We related serum levels of panIg against the S1/-receptor binding domain quantified by the un-adapted ECLIA with full virus neutralization capacity against SARS-CoV-2 B.1 or SARS-CoV-2 BA5.1. (3) Results: 92% of the sera exhibited sufficient neutralization capacity against the B.1 strain. Only 20% of the sera sufficiently inhibited the BA5.1 strain. Sera inhibiting BA5.1 could not be distinguished from non-inhibiting sera by serum levels of panIg against the S1/-receptor binding domain quantified by the un-adapted ECLIA. (4) Conclusion: Quantitative serological tests for an antibody against the S1/-receptor binding domain are unsuited as vaccination companion diagnostics, unless they are regularly adapted to mutations that have accumulated in that domain.

## 1. Introduction

The continued emergence of new SARS-CoV-2 mutants and the high incidence of re-infection and COVID-19 disease in vaccinated populations [1,2,3] clearly corroborate the need for regular re-vaccination, similar to vaccinations for influenza [4]. Initially, SARS-CoV-2 vaccinations followed fixed temporal schedules that were designed to break pandemic waves [5,6,7,8]. By now, health care deals with the outcome of a heterogeneous vaccination regimen and continuous asynchronous endemic re-infection by various SARS-CoV-2 mutants. As a consequence, immune responsiveness to SARS-CoV-2 exhibits considerable variability within the population [9], and individualization of vaccination has become an issue in several countries [10,11,12,13].

We [14] and others [10,11,15] have argued that diagnostic tests for humoral SARS-CoV-2 immune responses that are commercially available and practical in the setting of routine health care diagnostics [16,17,18] can be used to gauge humoral ex vivo immune responsiveness to SARS-CoV-2 and possibly provide a companion diagnostic for individualized re-vaccination. We observed that simple and rapid measurements of circulating SARS-CoV-2 antibody levels in serum were reasonably well correlated with virus-neutralizing activities determined by ex vivo surrogate assays or by the neutralization of the full SARS-CoV-2 virus in cell culture. We derived from these investigations a cut-off value for panIg antibodies against Spike S1-protein in serum. Above this cut-off value, immune responsiveness to SARS-CoV-2 could be assumed to be sufficient, as deduced from effective virus neutralization ex vivo [14].

However, these data were only valid for the original SARS-CoV-2 B.1 isolate and for vaccination and immune assays based on the unmodified spike S1-protein domain derived from the original virus strain. By now, the available serological assays are still the same and most persons still rely on protection by unmodified vaccines, but infectious challenges originate from mutant virus strains, in which the protein domain targeted by vaccines and putative companion diagnostics has accumulated many mutations and, thereby, is considerably altered [19]. Therefore, our published data [14] are probably superseded. To address that concern, we re-tested the post-vaccination sera of the above study with respect to their potency to neutralize the virus strain Omicron BA.5.1, which currently dominates the SARS-CoV-2-endemy in many countries [12,13]. 

## 2. Materials and Methods

Study features and most assays were previously described [14]. Initially, a total of 124 study participants (83 female, 41 male, mean age 46 years, median age 50 years) were recruited at the University Hospital of the Heinrich Heine University, Düsseldorf. All participants were employees of that institution and underwent a program of two vaccinations with the COVID-19 vaccine Spikevax (Moderna Biotech, Cambridge, MA, USA), spaced exactly four weeks apart. Vaccinations were performed according to the instructions of the manufacturer and the recommendations of the German vaccination commission (STIKO). None of the participants tested positive for SARS-CoV-2 or exhibited symptoms of COVID-19, nor did they exhibit debilitating symptoms of co-morbidities. We also retested 90 serum samples obtained six months after the second vaccinations (70 female, 20 male, mean age of 47 years, median age of 49 years) for their neutralization capacity against the SARS-CoV-2 Omicron variant BA5.1 (GISAID accession number EPI_ISL_16100719).

Antibodies against the SARS-CoV-2 spike (S1) protein receptor-binding domain (S1-AB) were determined using chemiluminescent immunoassay (ECLIA) (Elecsys Anti-SARS-CoV-2 S, Roche Diagnostics GmbH, Mannheim, Germany) on a COBAS 8000 analyzer (Roche, Basel, Switzerland), as prescribed by the manufacturer. Samples were measured at tenfold dilution (Roche Cobas Universal Diluent) and remeasured at 400-fold dilution when exceeding the upper detection limit (250 U/mL) [14].

The virus neutralization activity of the SARS-CoV-2 antibodies was measured with the surrogate assays NeutraLISA (EUROIMMUN Medizinische Labordiagnostika AG, Lübeck, Germany) and cPass(GenScript Biotech, Piscataway, NJ, USA), both of which measure the binding of the recombinant, biotin-labelled ACE2 receptor to the recombinant SARS-CoV-2-S1/-receptor-binding domain immobilized on microtiter plates [14].

Full-virus endpoint dilution neutralization (BA.5NT) was measured in duplicate in five-fold serial dilutions (1:10 to 1:1250) of heat-inactivated sera (56 °C, 30 min). A total of 10 µL of serum samples was incubated (37 °C, 1 h) with a SARS-CoV-2 Omicron BA5.1 virus solution at an absolute TCID50 of 100. Subsequently, 50 µL of cell suspension containing 25 × 10^4^ VeroE6 cells/mL (ATCC-CRL-1586) was added to each sample, and incubation continued (37 °C, 5% CO_2_, 96 h). Subsequently, cytopathic effects (CPEs) were determined by microscopic inspection. The effective neutralization titer was defined as the highest CPE-negative sample dilution. Titers of ≥1:10 were considered positive. Controls included in each test series encompassed neutralization-negative and -positive serum samples (previously determined and stored at −20 °C), the effect of virus in the absence of serum, and the growth controls of cells exposed to neither the virus nor the serum.

Graph Pad Prism 9 (Graph Pad Software Inc., San Diego, CA, USA) was used for analysis. Normal distribution was tested according to Shapiro–Wilk. Correlations were analyzed by Spearman correlation. Correlations were assumed to be good at r ≥ 0.7 and moderate at r ≥ 0.5. For all tests, statistical significance was assumed at *p* < 0.05. Missing data (about 12%) was handled by listwise deletion.

## 3. Results

The ex vivo immune responsiveness of the tested serum samples differed markedly between the B.1 and BA.5.1 strains. Of the previous samples, 92% (85/92) exhibited sufficient full-virus neutralization capacity against the B.1 strain [14]. In contrast, in this study, only 20% (18/90) of the samples exhibited sufficient full-virus neutralization capacity against the BA.5.1 strain (BA.5-NT). These differences were also apparent in quantitative comparisons: B1-NT titers exhibited reasonable correlations with two surrogate assays of virus neutralization that measure the inhibition of the binding of the recombinant, biotin-labelled ACE2-receptor to the recombinant SARS-CoV-2-S1/-receptor binding domain immobilized on microtiter plates (NeutraLISA, EUROIMMUN AG, Lübeck, Germany, and cPass, GenScript Biotech, Piscataway, NJ, USA). In contrast, BA.5-NT titers showed no quantitative correlations with these surrogate assays and were only moderately correlated with B.1-NT titers (Figure 1). 

In keeping with the data demonstrated in Figure 1, it was not possible to define cut-off values for the two surrogate assays that would allow a discrimination of BA.5-NT-positive samples. In fact, BA.5-NT-negative and -positive samples were completely intermingled relative to their capacity to inhibit the binding between the ACE2-receptor to the recombinant SARS-CoV-2-S1/-receptor binding domain (Figure 2a). Similarly, serum levels of panIg antibodies against the SARS-CoV-2 spike (S1) protein receptor binding domain (S1-AB) determined by chemiluminescent immunoassay (Roche Diagnostics, Mannheim, Germany) allowed, at best, a moderate discrimination of BA.5-NT-positive samples: 70% (12/17) of the samples above an S1-AB cut-off value of 1700 U/mL (Figure 2b, dotted line) were BA.5-NT-positive, while below that S1-AB value, 91% (67/73) of the samples were BA.5-NT-negative. We assume that this discriminatory power is not sufficient for diagnostic purposes.

## 4. Discussion

The salient findings of this study are:Vaccination with mRNA corresponding to the original sequence of the S1/-receptor binding domain (derived from SARS CoV-2 B.1) confers a much lower humoral ex vivo neutralizing potency against SARS CoV-2 Omicron B.A5.1 than against SARS CoV-2 B.1.Commercial serological tests based on the original S1/-receptor binding domain (derived from SARS CoV-2 B.1) have only limited predictive power for ex vivo neutralizing potency against SARS CoV-2 Omicron BA.5.1.

In summary, these observations are in line with published findings regarding the diminishing power of un-adapted SARS-CoV-2-vaccinations to protect against immune escape variants of the virus that have accumulated mutations in the S1/-receptor binding domain [12,13,19]. 

Our previous investigation addressed the immune responsiveness of post-vaccination sera against the SARS CoV-2 B.1 strain. We could derive from that investigation a diagnostic strategy mainly based on serum levels of panIg against the SARS-CoV-2 spike (S1) protein receptor binding domain, which provided a reliable way of gauging levels and functionality of circulating SARS-CoV-2 antibodies [14]. However, reanalysis of these samples with respect to the currently most abundant and clinically relevant mutant SARS CoV-2 BA.5.1 reveals that the above strategy is severely compromised by the immunological drift of the virus and can no longer be safely applied.

A limitation of our study is that our results refer only to the B.1 and B.5.1 variants studied here and not to other variants that have evolved in the meantime. Furthermore, since our collective was exclusively vaccinated with Spikevax (Moderna Biotech), our findings cannot be easily applied to other vaccines and additional investigations on later-developed protein- or vector-based vaccines would be necessary.

## 5. Conclusions

It is not astonishing that serological assays investigating antibody interactions with the S1/-receptor binding domain are compromised by mutations that accumulate in that domain and gradually lose their predictive power for humoral immune responsiveness to virus mutants such as SARS CoV-2 Omicron BA.5.1. We conclude that currently un-adapted serological assays are of low value as vaccination companion diagnostics, and that they must be adapted to the mutations of the clinically relevant virus strains, in a similar fashion to that of the vaccines themselves.

## Figures and Tables

**Figure 1 vaccines-11-00598-f001:**
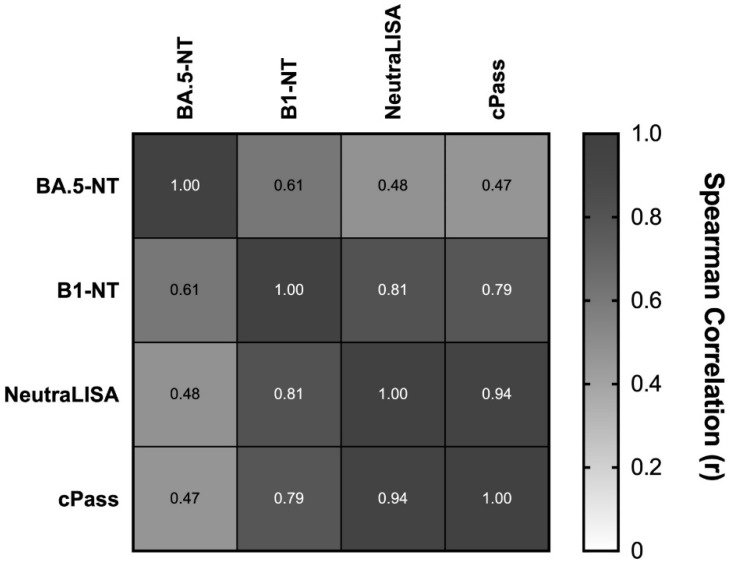
Comparison of virus neutralization capacity of 92 sera obtained six months after vaccination with the original mRNA vaccine (Spikevax, Moderna). Serological neutralization potency was determined by inhibition of the binding of the recombinant, biotin-labelled ACE2-receptor to the recombinant SARS-CoV-2-S1/-receptor binding domain (NeutraLISA, Euroimmun and cPass, Genscript Biotech) or by the endpoints of full-virus dilution neutralization using either the B.1-strain (B.1-NT) or the BA.5.1-strain (BA.5-NT) as targets. R-values of Spearman’s correlation test are shown.

**Figure 2 vaccines-11-00598-f002:**
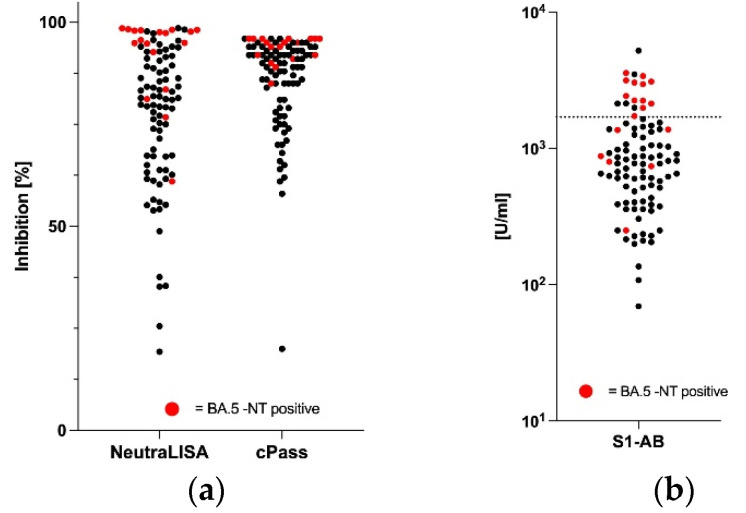
Discrimination of BA.5-NT-positive and -negative sera by quantitative serology. (**a**) Serological neutralization potency determined by inhibition of the binding of the recombinant, biotin-labelled ACE2-receptor to the recombinant SARS-CoV-2-S1/-receptor binding domain (cPass, Genscrip Biotech and NeutraLISA, Euroimmun); (**b**) serum levels of panIg against the SARS-CoV-2 spike (S1) protein receptor binding domain (S1-AB) determined by chemiluminescent immunoassay (ECLIA, Roche Diagnostics). Black dots: BA.5-NT-negative samples; red dots: BA.5-NT-positive samples; dotted line: cut-off at 1700 U/mL.

## Data Availability

Not applicable.

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
