# Peer review of "Ex Vivo Immune Responsiveness to SARS-CoV-2 Omicron BA.5.1 Following Vaccination with Unmodified mRNA-Vaccine"

_vaccines, 2023, doi:10.3390/vaccines11030598_

Round 1

Reviewer 1 Report

The authors retested serum samples, which were originally characterized for reactivity against the SARS-CoV2-2 Wuhan Hu-1 wildtype virus, for neutralization potency against the BA.5.1 variant. Correlations between the previously run tests and this new measurement were examined.

The observation of the communication is expectable, it strengthens other similar investigations that antigenic drift gradually impairs the predictive power of serological assays carried out with the original viral strain.

Remarks

The authors refer to their previous publication regarding the most assays. While detailed description is unnecessary, the name of those whose results are shown here could be listed under Mat&Met.

Correlation of the neutralization test BA.5.1 with the un-adapted ECLIA is not shown in the table. The ECLIA is important, being the most widely routinely available test, why is it left out?

The results suggest that serum samples with the strongest reactivity tend to be positive in the BA.5.1 neutralization test. The results of pseudo-neutralization tests and the ECLIA are non-linearly related to serum dilution. What were the dilutins used for the presented assays? Can serum dilutions be adjusted to improve correlation between their results and the BA.5.1 neutralization test?

Minor

Line 35 influenca : influenza

Author Response

REM 1. The authors refer to their previous publication regarding the most assays. While detailed description is unnecessary, the name of those whose results are shown here could be listed under Mat&Met

Reply: We were under the impression that according to good scientific practice it would suffice to cite the correct reference, when carrying out new analyses of published data that were obtained by a different team of investigators. Apparently, we were in error. Thus, we have explicitly acknowledged all contributors to our previous published investigation, which were not involved in the current one (line 204 ff). We suggest that the acknowledgment section would be the appropriate place to do this. Moreover, we would like to point out that all persons having contributed new data to the current study have been named as authors or coauthors in the manuscript.

REM 2. Correlation of the neutralization test BA.5.1 with the un-adapted ECLIA is not shown in the table. The ECLIA is important, being the most widely routinely available test, why is it left out?

Reply: Figure 1 is supposed to compare neutralizing capacities of serum samples concerning the two different VOCs. Correlation between spike antibodies measured with ECLIA are sufficiently represented in Figure 2b. The paper gives a new diagnostic algorithm and suggests a new cut-off for S1-AB of 1700 U/ml in the second part of the results while firstly neutralizing capacities as the strongest correlate to immune protection is analyzed.

REM 3. The results suggest that serum samples with the strongest reactivity tend to be positive in the BA.5.1 neutralization test. The results of pseudo-neutralization tests and the ECLIA are non-linearly related to serum dilution. What were the dilutions used for the presented assays? Can serum dilutions be adjusted to improve correlation between their results and the BA.5.1 neutralization test?

Reply: Samples were measured at 10-fold dilution (Roche Cobas Universal Diluent) and re-measured at 400-fold dilution when exceeding the upper detection limit (250 U/mL).
We do not expect that the dilution levels could be adjusted to obtain a better correlation between S1-AB and neutralization. Very few samples were even positive for the BA5 VOC in any of the tests. Improving the dilution levels would make sense in a highly sensitive area where methodical refinements would have a large impact on the results, but in our opinion this is not to be expected in a clear case as the one seen here.

Minor

Line 35 influenca : corrected into „influenza“

Reviewer 2 Report

The topic of this manuscript falls within the scope of Vaccines.

It is a quite interesting paper. The Authors showed that quantitative serological tests for antibody against the S1/-receptor binding domain are unsuited as vaccination companion diagnostics unless they are regularly adapted to mutations having  accumulated in that domain.

The Authors have presented sufficient data. The appropriate figures have been provided. The article is easy to read and logically structured.  The authors used appropriate statistical methods

There are only some  comments in the reviewer's opinion that should be taken under consideration by the Authors:

1.      Please add limitations of your study

2.      please separate the discussion section from the conclusion section

3.      Please add the section future trends of  testing

4.      In the section material and methods please add  inclusion and exclusion criteria

  • Please cite: DOI: 10.3390/jcm11030750, DOI: 10.1080/21645515.2022.2119766

Author Response

REM 1. Please add limitations of your study
Reply: We have included the limitations of our study under point 5 conclusions (line 171ff)

REM 2. Please separate the discussion section from the conclusion section
Reply: Done

REM 3. Please add the section future trends of testing

Reply: Done (line 184)

REM 4. In the section material and methods please add inclusion and exclusion criteria

Reply: We have included the limitations of our study (line 65ff)

REM 4b:  Please cite: DOI: 10.3390/jcm11030750, DOI: 10.1080/21645515.2022.2119766

Reply: We have appended these two references to the first § of the introduction (line 40/41) and the second § of the discussion (line 161). We hope the reviewer intended his papers to be cited in this context, since the bearing of these publications on the story at hand has not become entirely clear to us.